# HOX Transcript Antisense RNA *HOTAIR* Abrogates Vasculogenic Mimicry by Targeting the AngiomiR-204/FAK Axis in Triple Negative Breast Cancer Cells

**DOI:** 10.3390/ncrna6020019

**Published:** 2020-05-26

**Authors:** Allan Lozano-Romero, Horacio Astudillo-de la Vega, María Cruz del Rocío Terrones-Gurrola, Laurence A. Marchat, Daniel Hernández-Sotelo, Yarely M. Salinas-Vera, Rosalío Ramos-Payan, Macrina B. Silva-Cázares, Stephanie I. Nuñez-Olvera, Olga N. Hernández-de la Cruz, César López-Camarillo

**Affiliations:** 1Posgrado en Ciencias Genómicas, Universidad Autónoma de la Ciudad de México, Mexico City 03100, Mexico; lozallan@gmail.com (A.L.-R.); yarely.vera.ms@gmail.com (Y.M.S.-V.); blackvolkova@gmail.com (S.I.N.-O.); ediacara79@yahoo.com.mx (O.N.H.-d.l.C.); 2Laboratorio de Investigación en Cáncer y Terapia Celular, Hospital de Oncología, Centro Médico Nacional Siglo XXI, Mexico City 06720, Mexico; hastud2@aol.com; 3Coordinación Académica Región Altiplano, Universidad Autónoma de San Luis Potosí, San Luis Potosí 78760, Mexico; rocio.terrones@uaslp.mx (M.C.d.R.T.-G.); macrina.silva@uaslp.mx (M.B.S.-C.); 4Programa en Biomedicina Molecular y Red de Biotecnología, Instituto Politécnico Nacional, Mexico City 07320, Mexico; lmarchat@gmail.com; 5Laboratorio de Epigenética del Cáncer, Facultad de Ciencias Químico Biológicas, Chilpancingo de los Bravo 39000, Guerrero, Mexico; danhs1mx@yahoo.com; 6Facultad de Ciencias Químico Biológicas, Universidad Autónoma de Sinaloa, Culiacán 80040, Sinaloa, Mexico; rosaliorp@uas.edu.mx

**Keywords:** HOTAIR, miR-204, vasculogenic mimicry, cell migration, breast cancer

## Abstract

HOX transcript antisense RNA (HOTAIR) is an oncogenic long non-coding RNA frequently overexpressed in cancer. HOTAIR can enhance the malignant behavior of tumors by sponging microRNAs with tumor suppressor functions. Vasculogenic mimicry is a hypoxia-activated process in which tumor cells form three-dimensional (3D) channel-like networks, resembling endothelial blood vessels, to obtain nutrients. However, the role of HOTAIR in vasculogenic mimicry and the underlying mechanisms are unknown in human cancers. In the current study, we investigated the relevance of HOTAIR in hypoxia-induced vasculogenic mimicry in metastatic MDA-MB-231 and invasive Hs-578t triple negative breast cancer cells. Analysis of The Cancer Genome Atlas (TCGA) database using cBioPortal confirmed that HOTAIR was upregulated in clinical breast tumors relative to normal mammary tissues. Our quantitative RT-PCR assays showed a significant increase in HOTAIR levels after 48 h hypoxia relative to normoxia in breast cancer cell lines. Remarkably, knockdown of HOTAIR significantly abolished the hypoxia-induced vasculogenic mimicry which was accompanied by a reduction in the number of 3D channel-like networks and branch points. Likewise, HOTAIR silencing leads to reduced cell migration abilities of cancer cells. Bioinformatic analysis predicted that HOTAIR has a potential binding site for tumor suppressor miR-204. Luciferase reporter assays confirmed that HOTAIR is a competitive endogenous sponge of miR-204. Congruently, forced inhibition of HOTAIR in cells resulted in augmented miR-204 levels in breast cancer cells. Further bioinformatic analysis suggested that miR-204 can bind to the 3′ untranslated region of focal adhesion kinase 1 (FAK) transcript involved in cell migration. Western blot and luciferase reporter assays confirmed that FAK is a novel target of miR-204. Finally, silencing of HOTAIR resulted in low levels of cytoplasmic FAK protein and alterations in the organization of cellular cytoskeleton and focal adhesions. In summary, our results showed, for the first time, that HOTAIR mitigates cell migration and vasculogenic mimicry by targeting the miR-204/FAK axis in triple negative breast cancer cells.

## 1. Introduction

Long non-coding RNAs (lncRNAs) (also known as large intervening non-coding RNAs (lincRNAs)) are novel regulatory molecules which function mainly as scaffolds to promote protein and nucleic acids interactions that are required to regulate gene expression. HOX transcript antisense RNA (HOTAIR) is an oncogenic lncRNA frequently overexpressed in diverse types of human malignancies including breast cancer, colorectal cancer, and hepatocellular carcinoma [1]. Increased evidences have shown that HOTAIR controls key proteins and cellular pathways related to cell proliferation, migration, invasion, cancer stem cell-like differentiation, angiogenesis, and metastasis, while it inhibits apoptosis [2]. HOTAIR enhances the malignant behavior of tumors mainly by remodeling the chromatin state of promoter genes, and by sponging microRNAs with tumor suppressor functions leading to tumor progression [2,3]. Moreover, deregulated expression of HOTAIR in tumors has been associated with chemotherapy resistance and poor prognosis [4]. Notably, lncRNAs act as competitive endogenous RNAs (ceRNAs) of microRNAs. Indeed, recent studies have suggested that HOTAIR could sponge different microRNAs to promote tumorigenesis [5,6,7,8,9]. Nonetheless, the role of HOTAIR in the regulation of vasculogenic mimicry, a novel cancer hallmark which involves the formation of patterned three-dimensional (3D) channel-like networks by tumor cells, is unknown in human cancers.

Vasculogenic mimicry is a cellular process characterized by the ability of tumor cells to develop 3D channel networks, similar to endothelial vessels, which are functional as they contain a lumen with plasm, erythrocytes, and blood flow dynamics resembling blood vessels [10,11,12]. This mechanism challenges our conventional view that angiogenesis is the exclusive process by which tumors obtain nutrients, and it also can explain, in part, the anti-angiogenic therapy failures frequently observed in cancer patients [13]. Therefore, vasculogenic mimicry has been related to more aggressive clinical phenotypes and increased risk of metastasis in solid tumors [14,15]. Remarkably, tumor microenvironment hypoxia is a pivotal factor triggering vasculogenic mimicry. Nevertheless, the function of lncRNA HOTAIR in hypoxia-induced vasculogenic mimicry and the underlying mechanisms are understood in breast cancer. Here, our main objective was to study lncRNAs regulating vasculogenic mimicry. As we learned that HOTAIR is also regulated by hypoxia, it was understandable to hypothesize that it also could be regulating vasculogenic mimicry. To the best of our knowledge, these findings have not been reported for HOTAIR, and thus the data are novel. In the present study, we investigated the role of HOTAIR in cell migration and hypoxia-induced vasculogenic mimicry in triple negative breast cancer cells.

## 2. Results

### 2.1. HOTAIR Expression Is Increased Under Hypoxia in Breast Cancer Cells

To initiate the study of HOTAIR in breast cancer cells, we first analyzed its expression using data from The Cancer Genome Atlas (TCGA) database using cBioPortal. The results confirmed that HOTAIR expression was significantly upregulated in triple negative breast tumors relative to normal mammary tissues (Figure 1A). Because vasculogenic mimicry can be induced by hypoxia, the HOTAIR expression was further analyzed in cells grown under low oxygen (1% O_2_) conditions for 48 h. The data from quantitative RT-PCR assays showed a significant increase (*p* < 0.05) in HOTAIR levels after hypoxia as compared with normoxia conditions in metastatic MDA-MB-231 and invasive Hs-578t triple negative breast cancer cells (Figure 1B).

### 2.2. Inhibition of HOTAIR Impairs Hypoxia-Induced Vasculogenic Mimicry in Breast Cancer Cells

To investigate the functional role of HOTAIR in vasculogenic mimicry, we performed knockdown of HOTAIR expression using a specific validated siRNA (hereafter named as DsiRNA HOTAIR inhibitor). The data showed that HOTAIR expression was significantly (*p* < 0.05) downregulated in DsiRNA HOTAIR inhibitor (30 nM) transfected cells as compared with non-transfected control and scrambled control in MDA-MB-231 (Figure 1C,D) and Hs-578t (Figure 1C,F) cells growing under normoxia and hypoxia conditions. Then, we evaluated the effect of HOTAIR silencing in 3D channel-like formation, representative of the early stages of vasculogenic mimicry. MDA-MB-231 and Hs-578t cells were grown under hypoxia for 48 h, placed over matrigel, and incubated for a further 0, 3, 6, and 9 h to track the development of 3D capillary-like networks. Our results showed that the non-transfected (Figure 2A, left panel) and scrambled control (Figure 2A, middle panel) MDA-MB-231 control cells exhibited vasculogenic mimicry-related networks after 3 h and 6 h grown in matrigel, which were increased at 9 h. Interestingly, knockdown of HOTAIR resulted in a dramatic inhibition of 3D cell networks (Figure 2A, right panel). A significant reduction in the number of branch points and capillary-like tubes (up to 95%) as compared with non-treated and scramble transfected control cells was found (Figure 2B,C). There were no changes in cell viability found at the tested concentration of DsiRNA HOTAIR inhibitor, indicating the impact in vasculogenic mimicry inhibition was not due to unspecific effects in cell survival of cancer cells. To confirm these findings, we also analyzed the metastatic and triple negative Hs-578t breast cancer cells. The results showed that the Hs-578t cells efficiently developed hypoxia-induced 3D channels formation after 6–9 h of incubation on matrigel (Figure 2D). Likewise, silencing of HOTAIR leads to a significant inhibition of vasculogenic mimicry as compared with the controls, although to a less extent than found in MDA-MB-231 cells (Figure 2D, right panel). A significant reduction in the number of branch points and capillary-like tubes as compared with the non-treated and scramble transfected control cells was observed (Figure 2E,F).

### 2.3. HOTAIR Regulates Cell Migration in Breast Cancer Cells

Cell migration is essential for vasculogenic mimicry as cells should proliferate, migrate, and align to form the 3D channel-like structures in response to hypoxia [16,17]. Therefore, we asked if HOTAIR could regulate vasculogenic mimicry, at least in part, by controlling cell migration. To test this hypothesis, we performed scratch/wound healing assays in breast cancer cells. The results showed that the silencing of HOTAIR resulted in a significant inhibition of migration abilities in MDA-MB-231 cells as compared with the non-transfected and scramble transfected control cells (Figure 3A,B). A similar significant inhibitory effect of DsiRNA HOTAIR inhibition on cell migration was also observed in Hs-578t cells (Figure 3C,D).

### 2.4. HOTAIR Sponges the Tumor Suppressor miR-204

Oncogenic lncRNAs can hijack microRNAs with tumor suppressor functions. We previously reported that miR-204 is a tumor suppressor microRNA with functions in angiogenesis, cell migration, and vasculogenic mimicry which is frequently downregulated in breast tumors and cell lines [18,19]. Therefore, we hypothesized that HOTAIR can inhibit vasculogenic mimicry and cell migration, at least in part, by sequestering miR-204. To test this possibility, we searched for potential miR-204-binding sites in HOTAIR sequence. The data showed that HOTAIR contains a conserved potential binding site for miR-204 at 131–145 nucleotide position (Figure 4A), which was selected and cloned downstream of luciferase reporter gene in pmiR-Report vector (named as p-miR-HOTAIR wt) and submitted to luciferase activity assays. A mutated version of miR-204 binding site in HOTAIR sequence was included as the control (dubbed as p-miR-HOTAIR mut). The results showed that after co-transfection of pmiR-HOTAIR wt construct plus miR-204 mimics, a diminished luciferase activity was found as compared with the following: (i) p-miR-HOTAIR wt alone and (ii) p-miR-HOTAIR wt plus scramble transfected control cells (Figure 4B). The mutated p-miR-HOTAIR mut construct did not show effects in luciferase activity indicating that the binding of HOTAIR to miR-204 was sequence specific (Figure 4B). To confirm these assumptions, we analyzed the miR-204 levels in cells treated with DsiRNA HOTAIR inhibitor by quantitative RT-PCR assays. The data showed that miR-204 levels were increased in HOTAIR-deficient MDA-MB-231 and Hs-578t cancer cells. All together, these data indicate that HOTAIR sponges miR-204 in breast cancer cells.

### 2.5. FAK Is a Novel Target of miR-204 in Breast Cancer Cells

Our computational analysis showed that the focal adhesion kinase 1 (FAK, also knowns as TK2 protein tyrosine kinase 2 PTK2) transcript contains a potential miR-204 binding site at 3′UTR (1507–1513 nucleotide position). We selected FAK for further analysis for the following two reasons: (i) our previous studies using phosphoproteomic profiling showed that miR-204 overexpression resulted in reduced expression/phosphorylation of FAK (and other kinases) which was associated with vasculogenic mimicry inhibition in two triple negative breast cancer cell lines [18,19] and (ii) FAK is a well-known protein involved in cell migration, which is essential for the development of channel structures during vasculogenic mimicry. Thus, to investigate if miR-204 can bind to the 3′UTR and downregulate the expression of FAK, a DNA fragment corresponding to 3′UTR of the FAK gene was inserted downstream of the luciferase gene into the pmiR-report vector (Figure 5A). A mutated version of the miR-204 binding site was used as the control. Then, the resulting wild type and mutated pmiR-LUC-FAK constructs were transfected together, and miR-204 mimics into MDA-MB-231 cells and luciferase activities in total protein extracts were analyzed after 24 h. The results indicated that ectopic co-transfection of miR-204 mimics and recombinant pmiR-LUC-FAK plasmid produced a significant reduction of the relative luciferase activity as compared with the non-transfected and scramble transfected controls (Figure 5B). The mutated sequence of FAK 3′UTR did not produced significant changes in luciferase activity, suggesting that miR-204 binding was specific.

To confirm the targeting of FAK by miR-204 we performed Western blot and immunofluorescence assays using specific antibodies. The results from immunoblotting showed that the FAK protein was expressed in MDA-MB-231 cells, but it was significantly (*p* < 0.01) decreased in cells transfected with miR-204 mimics as compared with the non-treated and scramble transfected controls cells (Figure 5C). There were no significant changes observed in GADPH protein levels used as the control. Likewise, immunofluorescence assays showed abundant FAK protein mainly located in the cytoplasm and accumulated in discrete focal points at the cell membrane (Figure 5D). Interestingly, a dramatic decrease in abundance of cytoplasmic FAK was observed in cells transfected with DsiRNA HOTAIR inhibitor as compared with the non-treated control cells (Figure 5D,E). These changes in FAK levels were accompanied by alterations in the localization of FAK and changes in the organization of cellular cytoskeleton. All together, these data confirmed that FAK is a novel target of miR-204.

## 3. Discussion

A limited number of studies have reported that lncRNAs can modulate the cellular processes associated with vasculogenic mimicry in cancer. For instance, an expression profiling of lncRNAs deregulated in aggressive osteosarcoma cells identified novel lncRNAs associated with tumorigenesis and vasculogenic mimicry. One of the most deregulated lncRNAs found in osteosarcoma cells was n340532. Knockdown of n340532 reduced 3D channel formation and showed potent anti-metastasis effects in vivo [20]. Likewise, Shi and coworkers reported that knockdown of lncRNA AFAP1-AS1 inhibited tumorigenesis, epithelial-mesenchymal transition, and vasculogenic mimicry by sponging RhoC and altering the ROCK1/p38MAPK/Twist1axys in osteosarcoma cells [21]. In addition, it was found that lncRNA MALAT1 expression was associated with the presence of vasculogenic mimicry and endothelial vessels in gastric cancer tissues. In vitro MALAT1 knockdown resulted in reduced cell migration, invasion, metastasis, and vasculogenic mimicry [22]. In this context, recent studies have suggested the role of lncRNAs in tumorigenesis and in the regulation of processes related to vasculogenic mimicry [23,24,25,26,27,28]. However, the role of HOTAIR in vasculogenic mimicry has not been studied in human cancers. HOTAIR was initially described as an epigenetic factor that functions in chromosomal remodeling and coordinates the recruiting of polycomb repressive complex 2 to gene promoters regulating in this way the cancer epigenome [3]. HOTAIR is transcribed from the genomic HOXC locus and represses expression in the distal HOXD locus and genes located on other genomic *loci*, resulting in decreased expression of multiple genes to promote metastasis of breast tumors [3]. Other molecular mechanisms involved in lncRNA-mediated effects on tumorigenesis are based on the interactions between lncRNAs and microRNAs, which is known as sponge effect, resulting in the derepression of microRNAs gene targets.

Here, we showed that HOTAIR has a pivotal role in hypoxia-induced vasculogenic mimicry in metastatic MDA-MB-231 and invasive Hs-578t triple negative breast cancer cells. First, we found that HOTAIR levels were significantly increased after 48 h hypoxia relative to normoxia in breast cancer cell lines, indicating a functional link between hypoxia and HOTAIR functions. These data are in agreement with an early report showing that HOTAIR promoter region carried out hypoxia-responsive elements (HREs) which are targeted by HIF-1α protein in non-small cells lung cancer cells. Moreover, HIF-1α knockdown inhibited HOTAIR upregulation under hypoxic conditions [29]. We have previously demonstrated that under hypoxic conditions HIF-1α expression was downregulated by miR-204 mimics in breast cancer cells [18] which could explain, in part, the negative effects of miR-204 we found in hypoxia-induced vasculogenic mimicry associated with HOTAIR upregulation. These observations are in line with our findings that knockdown of HOTAIR impaired the hypoxia-induced vasculogenic mimicry which was accompanied by a reduced cell migration of breast cancer cells. Moreover, in this investigation, we confirmed that HOTAIR is a competitive endogenous sponge of miR-204, which was congruent with our findings showing that the inhibition of HOTAIR resulted in a modest but significant increase in miR-204 levels after transfection of a synthetic dsiRNA HOTAIR inhibitor. These data suggest that higher dsiRNA inhibitor concentration could be needed to achieve the complete derepression of miR-204. However, it was enough to exert a significant effect on the vasculogenic mimicry ability of cancer cells, confirming the important role of this non-coding RNA in the process. In addition, this could reflect the existence of additional mechanisms controlling the miR-204 abundance in cells operating together with HOTAIR sponge function. Moreover, it suggests that forced downregulation of HOTAIR with DsiRNAs can result in the alterations of other bait proteins and miRNAs operating in vasculogenic mimicry, which were not identified here. Finally, using diverse experimental approaches, we showed that FAK is a novel target of miR-204, and that silencing of HOTAIR resulted in low levels of FAK protein and alterations in the organization of cellular cytoskeleton and focal adhesions, associated with the derepression of miR-204. Remarkably, previous immunohistochemical analysis of the FAK protein in triple negative breast tumors have correlated high FAK levels with shorter overall survival and progression-free survival in patients with metastatic tumors [30], suggesting that FAK functions in tumor cell migration and metastasis are relevant to invasive disease progression. Likewise, another report showed that high FAK protein expression was significantly correlated with aggressive HER2 and triple negative subtypes, presence of lymph node metastases, distant metastasis, and with shortened recurrence free and overall survival rates [31]. On the other hand, the clinical significance of HOTAIR/miR-204 axis has been previously reported by us [18,19] and other researchers [32,33]. HOTAIR is oncogenic and miR-204 is a tumor suppressor, accordingly they showed high and low expression in breast carcinoma, respectively, and dictate low overall survival and poor prognosis. For instance, a study showed that overexpression of HOTAIR in breast carcinoma was not associated with nodal metastases or prognosis in ER-positive patients, however, it predicted a poor prognostic in ER-negative patients with node-positive [32]. In contrast, Sorensen and coworkers showed that high HOTAIR expression in primary tumors was significantly associated with worse prognosis independent of prognostic markers mainly in estrogen receptor (ER)-positive tumors [33]. These data indicate that interplay and clinical value of HOTAIR/miR-204/FAK axis in breast cancer subtypes predicts worst overall survival, however, it needs to be further confirmed in large cohort studies. In summary, our results suggest that HOTAIR inhibits cell migration and vasculogenic mimicry by targeting the miR-204/FAK axis in triple negative breast cancer cells. These data suggest that suppression of HOTAIR upon hypoxia of breast cancer cells could be a novel therapeutic strategy.

## 4. Materials and Methods

### 4.1. Cell Lines

Human breast carcinoma MDA-MB-231 and Hs-578t triple negative breast cancer cell lines were obtained from the American Type Culture Collection and maintained in Dulbecco’s modification of Eagle’s minimal medium (DMEM) supplemented with 10% fetal bovine serum, and penicillin-streptomycin (50 unit/mL; Invitrogen, Carlsbad, CA, USA).

### 4.2. Transfection of DsiRNA HOTAIR Inhibitor and Precursor miR-204

MiR-204 mimics precursor (AM17110; Thermo Fisher Scientific Inc., Waltham, MA, USA) and pre-miR-negative scrambled control (AM17110; Thermo Fisher Scientific) were transfected into MDA-MB-231 and Hs-578t cells using siPORT amine transfection agent (Thermo Fisher Scientific). DsiRNA HOTAIR inhibitor (30 nM, IDT Technologies ID M227532134) was transfected in cells following the same experimental procedure as that of miR-204 and control. Briefly, DsiRNA HOTAIR or miR-204 mimics and scramble control (30 nM) were individually added to wells containing 1 × 10^7^ cells cultured in DMEM for 48 h. Overexpression of miR-204 and silencing of HOTAIR were confirmed by quantitative RT-PCR at 48 h post-transfection using total RNA. MiR-204-expressing cells and HOTAIR-deficient cells were used for downstream analysis.

### 4.3. Vasculogenic Mimicry Inhibition Assays

Vasculogenic mimicry experiments were performed through 3D cultures on matrigel. MDA-MB-231 and Hs-578t cells (1 × 10^4^ cells/well) were transfected with miR-204 mimics (30 nM) or scramble (30 nM) negative control, as previously described [18]. Afterward, cells were cultured in a 96-well plate covered with 50 µL matrigel geltrex matrix. Then, cells were incubated at 37 °C in 5% CO_2_ atmosphere under normoxia or hypoxia conditions (1% O_2_), for 48 h. Formation of 3D channel-like networks which are representative of the initial stages of vasculogenic mimicry were quantified by counting under an inverted microscope (Iroscope SI-PH) for 0, 6, and 12 h. Two observers individually counted the number of branch points and tubular-like structures. Data were expressed as mean ±S.D. A *p* < 0.05 was considered as statistically significant.

### 4.4. RNA Isolation

Total RNA from MDA-MB-231 and Hs-578t cells was extracted using 500 µL Trizol for 1 × 10^4^ cells/well, as described by the manufacturer (Thermo Fisher Scientific). Total RNA integrity and concentration were assessed using a capillary electrophoresis system (Agilent 2100 Bioanalyzer) before downstream processing.

### 4.5. Bioinformatics Analysis

MiR-204 target genes were predicted using TargetScan 7.0 (http://www.targetscan.org/vert_71/) and PicTar (http://www.pictar.org/) softwares. Only target genes that were predicted by the two algorithms were selected for further analysis. Searching of miR-204 binding sites in HOTAIR RNA sequence was performed using (http://zmf.umm.uni-heidelberg.de/apps/zmf/mirwalk2/) miRWalk v.2.0 (http://mirwalk.umm.uni-heidelberg.de/). HOTAIR expression data was obtained from TCGA datasets using cBioPortal (http://www.cbioportal.org/).

### 4.6. Western Blot Assays

30 μg of whole protein extracts obtained from breast cancer cells were separated on 12% SDS-PAGE and transferred to 0.2 μm nitrocellulose membrane (Bio-Rad Laboratories, Hercules, CA, USA), and then incubated with the following primary antibodies: anti-FAK (dilution 1:1000; Cell Signaling Technology Inc., Danvers, MA, USA) and anti-GAPDH (dilution 1:1000; sc-365062; Santa Cruz Biotechnology, Santa Cruz, CA, USA). Immunodetected bands were quantified by densitometry analysis using the myImage Analysis software (http://lklakassd).

### 4.7. Luciferase Gene Reporter Assays

To define if miR-204 targets the FAK gene, a DNA fragment of FAK 3′UTR containing the predicted miR-204 binding site wild type and mutated were cloned into p-miR-report vector (Thermo Fisher Scientific) into the *Bam*H1*/Sac*1 restriction sites downstream of luciferase gene. The recombinant pmiR-LUC-FAK wt and pmiR-LUC-FAK mut plasmids were purified from bacteria and verified through *Bam*H1/*Sac*1 enzymatic restriction and automatic sequencing. Then, pmiR-LUC and pmiR-LUC-FAK plasmids were transfected into MDA-MB-231 cells using a lipofectamine 2000 (Thermo Fisher Scientific). At 24 h after transfection, pre-miR-204 (30 nM) and scramble (30 nM) were co-transfected with lipofectamine RNAi max (Thermo Fisher Scientific), and then 24 h after cotransfection, the firefly and *Renilla reniformis* luciferase activities were measured using the Dual-Glo luciferase Assay (Promega, Madison, WI, USA) using a Fluoroskan Ascent™ Microplate Fluorometer. Firefly luciferase activities were normalized with *Renilla reniformis* luciferase.

### 4.8. Immunofluorescence Analysis

Briefly, cells were fixed in 4% formaldehyde in PBS 1X for 30 min at room temperature. Coverslips were incubated with 0.1% Triton X-100 for 3 min. Following washing with PBS 1X, cells were blocked for 40 min at room temperature with 0.2% BSA in PBS 1X, and incubated with Phalloidin 1X (ab235138, Abcam, Cambridge, UK) for 30 min at room temperature with anti-FAK antibodies overnight at 4 °C. Then, stained cells were washed with PBS 1X for 15 min and mounted for confocal microscopy.

### 4.9. Statistical Analysis

Experiments were performed three times by triplicate and results were represented as mean ± S.D. One-way analysis of variance (ANOVA) was used to compare the differences between means. A *p* < 0.05 was considered as statistically significant.

## 5. Conclusions

Here, we provide experimental evidences suggesting that HOTAIR has a relevant function in hypoxia-induced vasculogenic mimicry and cell migration in breast cancer cells. We demonstrated that HOTAIR regulatory effects are hypoxia dependent, confirming early findings suggesting that HOTAIR is responsive to low oxygen conditions. In addition, we evidenced that focal adhesion kinase 1 (FAK) transcript is a novel target of miR-204-HOTAIR axis whose intervention resulted in low levels of cytoplasmic FAK protein and alterations in the organization of cellular cytoskeleton and focal adhesions. In conclusion, our data suggested that HOTAIR inhibits cell migration and vasculogenic mimicry by targeting the miR-204/FAK axis in triple negative breast cancer cells.

## Figures and Tables

**Figure 1 ncrna-06-00019-f001:**
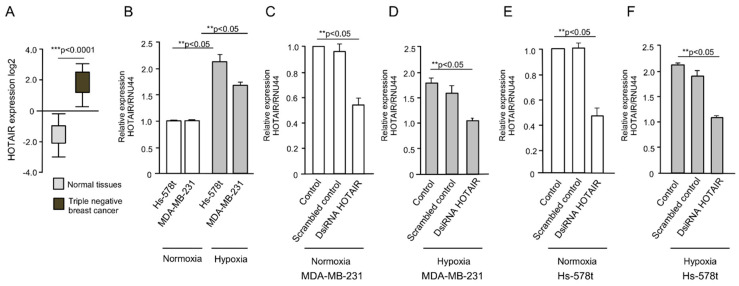
Expression of HOX transcript antisense RNA (HOTAIR) in breast tumors and HOTAIR silencing in breast cancer cell lines. (**A**) Expression of HOTAIR in healthy tissues (*n* = 99) and triple negative breast tumors (*n* = 217) from The Cancer Genome Atlas (TCGA) data using cBioPortal; (**B**) qRT-PCR assays showing the expression of HOTAIR in MDA-MB-231 and Hs-578t cells in normoxia and hypoxia conditions; (**C**–**F**) qRT-PCR assays showing the HOTAIR levels in HOTAIR-silenced cells using the DsiRNA HOTAIR inhibitor (35 nM) in normoxia and hypoxia conditions in (**C**,**D**) MDA-MB-231, and (**E**,**F**) Hs-578t cells. RNU44 was used as endogenous normalizer. Assays were performed three times by triplicate and data were expressed as mean ± S.D. ** *p* < 0.05; *** *p* < 0.0001.

**Figure 2 ncrna-06-00019-f002:**
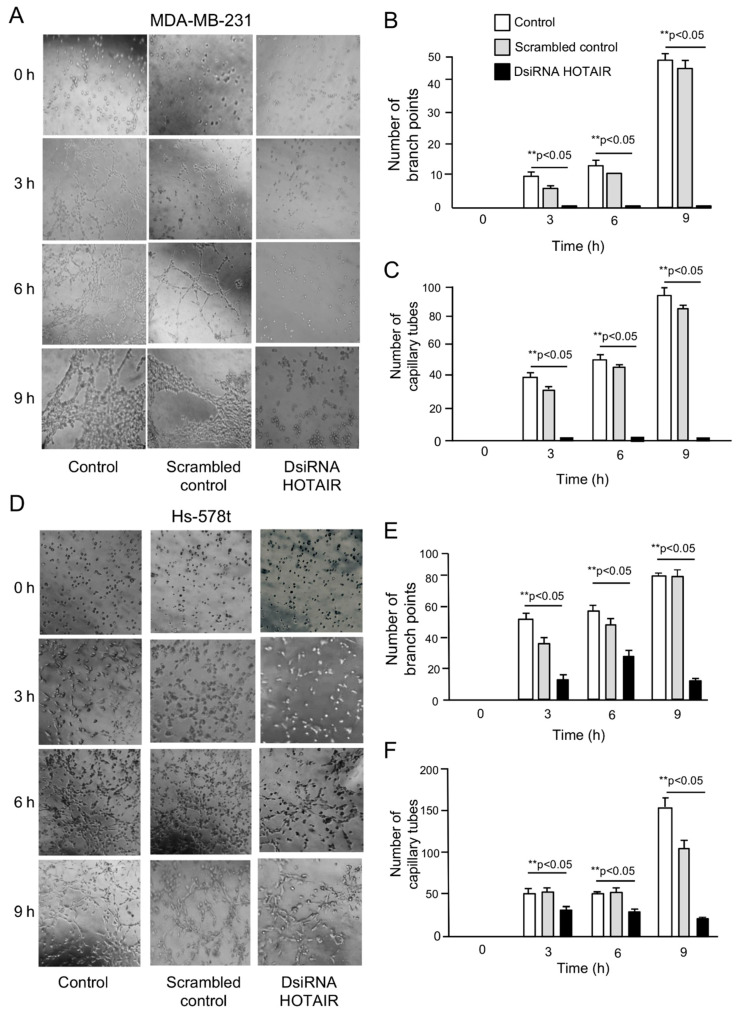
Knockdown of HOTAIR inhibits vasculogenic mimicry. After 48 h of incubation in hypoxia, breast cancer cells were placed on matrigel, and then imaged at the indicated times (0–9 h). (**A**) Vasculogenic mimicry images of MDA-MB-231 cells treated with DsiRNA HOTAIR inhibitor (right panel), scrambled (middle panel), and non-transfected control (left panel) conditions; (**B**,**C**) Graphical representation of the number of branch points and capillary tubes from panel A; (**D**) Vasculogenic mimicry images of Hs-578t cells treated with DsiRNA HOTAIR inhibitor (right panel), scrambled (middle panel), and non-transfected control (left panel) conditions; (**E**,**F**) Graphical representation of the number of branch points and capillary tubes from panel D. Experiments were performed three times by triplicate and data were expressed as mean ± S.D. ***p* < 0.05.

**Figure 3 ncrna-06-00019-f003:**
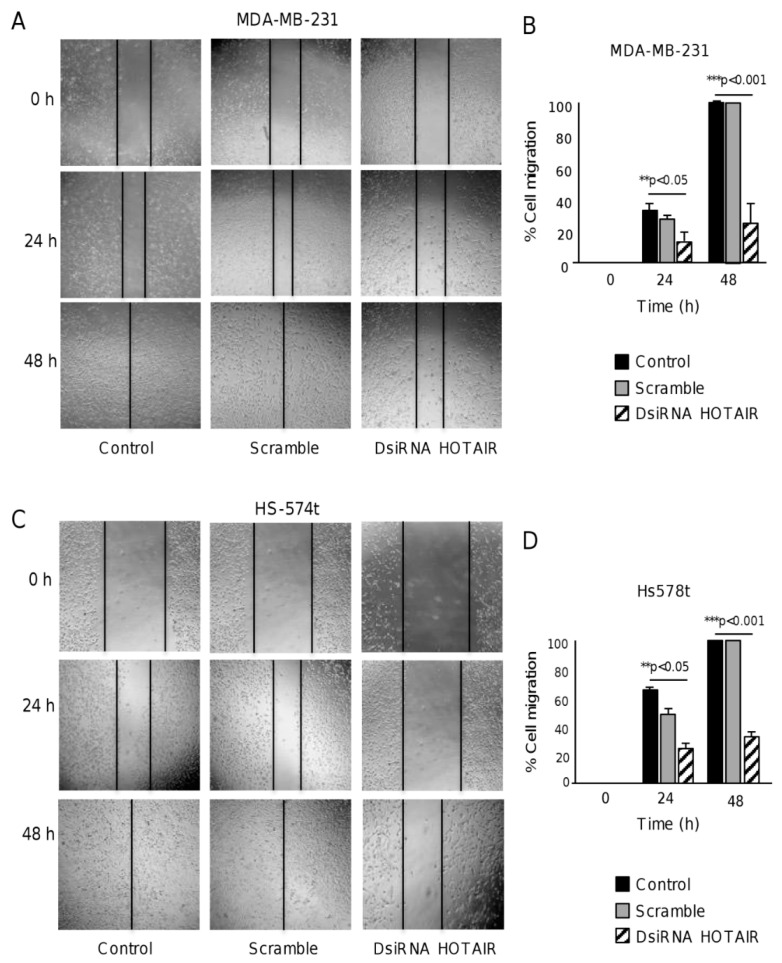
The inhibition of HOTAIR decreases cell migration in breast cancer cells. Scratch/wound-healing assays in (**A**,**B**) MDA-MB-231 and (**C**,**D**) Hs-578t cancer cells treated with DsiRNA HOTAIR inhibitor (right panel), scramble (middle panel), and non-transfected control (left panel). Cell migration was documented for 0, 24, and 48 h. The graphical representation of the migration is shown in (**B**) for MDA-MB-231 and in (**D**) for Hs-578t. Experiments were performed three times by triplicate and data were expressed as mean ± S.D. ** *p* < 0.05 and *** *p* < 0.001.

**Figure 4 ncrna-06-00019-f004:**
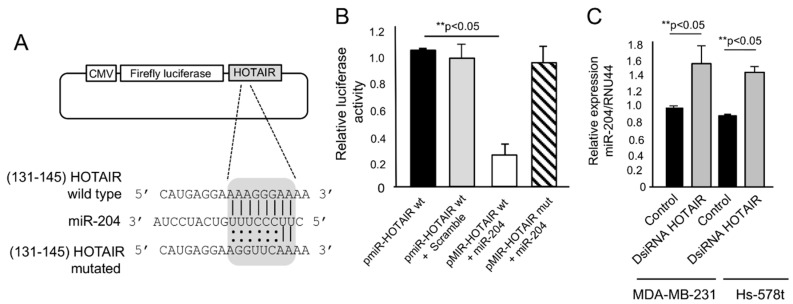
HOTAIR acts as an endogenous competitive RNA sponge for miR-204. (**A**) pmiR-LUC-HOTAIR constructions showing the position of HOTAIR wild type and mutated sequences, and the containing seed region for miR-204 in grey box; (**B**) Firefly luciferase activity assays in MDA-MB-231 cells co-transfected with pmiR-LUC-HOTAIR wild type and mutated plasmids and miR-204 mimics; (**C**) qRT-PCR assays for miR-204 in MDA-MB-231 and Hs-578t cells transfected with DsiRNA HOTAIR inhibitor and untreated control. Experiments were performed three times by triplicate and data were expressed as mean ± S.D. ** *p* < 0.05.

**Figure 5 ncrna-06-00019-f005:**
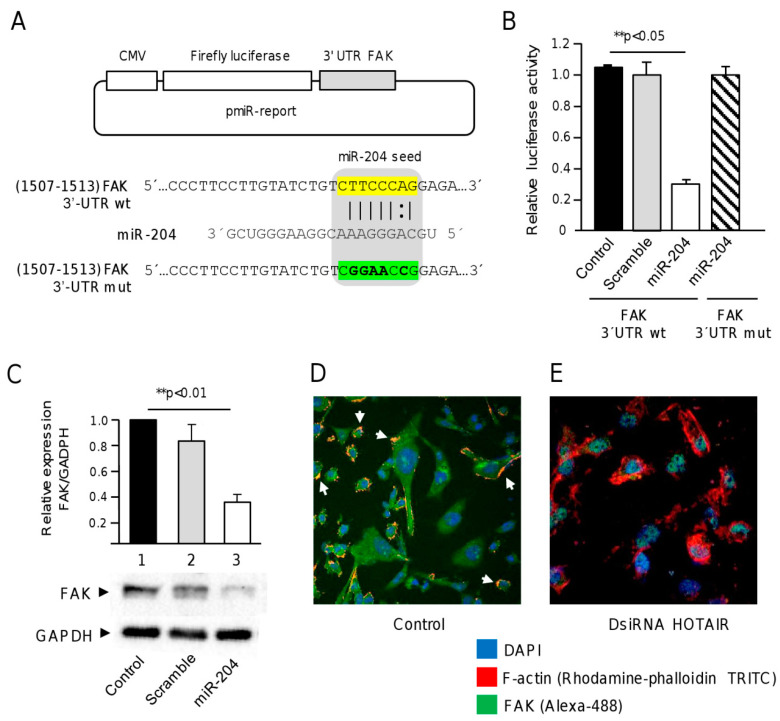
FAK is a novel target of miR-204. (**A**) pmiR-LUC-FAK plasmids showing the seed region for miR-204 in the 3′UTR of wild type and mutated FAK sequences; (**B**) Firefly luciferase activity assays in MDA-MB-231 cells co-transfected with wild type and mutated pmiR-LUC-FAK plasmids and miR-204 mimics; (**C**) Western blot assays FAK protein in MDA-MB-231 cells treated with miR-204 mimics (lane 3), scramble (lane 2), and non-transfected control (lane 1); (**D**,**E**) Immunofluorescence and confocal microscopy assays using anti-FAK antibodies (Alexa-488, green channel), rhodamine-phalloidin (TRITC, red channel) for cytoskeleton staining, and DAPI for nuclear DNA staining (blue channel) in MDA-MB-231 cells treated with (**E**) DsiRNA HOTAIR and (**D**) non-transfected control. Experiments were performed three times by triplicate and data were expressed as mean ± S.D. ** *p* < 0.05.

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
