# Peer review of "HOX Transcript Antisense RNA HOTAIR Abrogates Vasculogenic Mimicry by Targeting the AngiomiR-204/FAK Axis in Triple Negative Breast Cancer Cells"

_ncrna, 2020, doi:10.3390/ncrna6020019_

Round 1

Reviewer 1 Report

In this paper, Lonzano-Romero and colleagues, study the role of HOTAIR in vasculogenic mimicry in hypoxia conditions. The authors have shown that HOTAIR KD significantly reduces this phenomenon. They went on showing that this happens via mir-204 regulation of FAK. Results are convincing and the authors' claims are supported by data.

Minor things to be addressed in the new version of the manuscript:

Fig. 2C, please comment on the effect of the scrambled control on the number of capillaries. Images show very small changes vs showed graphs.

Fig. 4 C, the increase of mir-204 is very modest, please comment.

Fig. 1A, increase scale on the Y axis panel B, the control is scrambled control, not scramble.

Author Response

Reviewer 1

In this paper, Lonzano-Romero and colleagues, study the role of HOTAIR in vasculogenic mimicry in hypoxia conditions. The authors have shown that HOTAIR KD significantly reduces this phenomenon. They went on showing that this happens via mir-204 regulation of FAK. Results are convincing and the authors' claims are supported by data.

Reply: We acknowledge to editors and reviewers 1 and 2 for the opportunity to revise the manuscript. Your critical suggestions that we have fully replied increase the quality of our study. All amendments have been marked in yellow color in the revised text (ncrna-794667 with highlighted changes.doc) for your easy reference and reading. We have carefully reviewed the manuscript according the referee suggestions and provide a point-by-point response.

Minor things to be addressed in the new version of the manuscript:

Fig. 2C, please comment on the effect of the scrambled control on the number of capillaries. Images show very small changes vs showed graphs.

Reply: Thank you very much for the wise comments. We agree with author observation about the discrepancy in the number of capillary observed in the images and graphs. We have double checked and quantified again the number of capillaries in replicates images. Data from this new analysis of images was graphed for scrambled control in figure 2C specifically at 3 and 6 h confirming minimal changes in capillary tube number in cells transfected with scramble relative to non-treated control. No significant changes of p-values were found.

Fig. 4 C, the increase of mir-204 is very modest, please comment.

Reply: Thanks you for the comment. We agree with reviewer that transfection of the DsiRNA HOTAIR resulted in a modest increase in miR-204 levels suggesting that higher inhibitor concentration may be needed to achieve the complete derepression of miR-204. However, it was enough to exert a significant effect in the vasculogenic mimicry ability of cancer cells confirming the important role of this non-coding RNA in the process. This also may reflect the existence of additional mechanisms controlling the miR-204 abundance in cells operating together with HOTAIR sponge function. Moreover, it suggests that forced downregulation of HOTAIR with DsiRNAs can results in the alterations of other genes and mechanisms operating in vasculogenic mimicry which were not identified here. These ideas have been now added to discussion section page 8, line 253-261.

Fig. 1A, increase scale on the Y axis panel B, the control is scrambled control, not scramble.

Reply: We have increased the scale on the Y-axis in figure 1B and indicated scrambled control as reviewer suggested.

Reviewer 2 Report

A nice paper which shows HOTAIR mitigates cell migration and vasculogenic mimicry by targeting the miR-204/FAK axis in triple negative breast cancer cells. Results are clearly presented and the conclusions drawn are appropriate given the results and data found.

Author Response

Reviewer 2

A nice paper which shows HOTAIR mitigates cell migration and vasculogenic mimicry by targeting the miR-204/FAK axis in triple negative breast cancer cells. Results are clearly presented and the conclusions drawn are appropriate given the results and data found.

Reply: Thanks you very much for the positive comments.

Reviewer 3 Report

In this manuscript the authors report the role of HOTAIR and mir-204 under  triple negative breast cancer. The manuscript describes   interesting observations related to the effects on vascular mimicry and invasion.

However, the authors should address the following points to improve the quality of the manuscript.

1) The authors should indicate which patients they analysed in Figure 1. Are those data from triple negative tumour samples? This is important as they claim their mechanism of action operates in triple negative cancer.

2) The authors should correlate their finding from two cell lines to clinical parameters using the TCGA dataset. For instance, what is the correlation of the studies transcripts/proteins to stage of disease, survival? Again, this should  be placed in the context of triple negative breast cancer patients.

2) Check Figure 2. One of the panel has a different colour background.

Round 2

Reviewer 3 Report

The authors have addressed the requested revision